# A Head-to-Head Comparative Study of the Replication-Competent Vaccinia Virus and AAV1-Based Malaria Vaccine versus RTS,S/AS01 in Murine Models

**DOI:** 10.3390/vaccines12101155

**Published:** 2024-10-10

**Authors:** Kartika Hardianti Zainal, Ammar Abdurrahman Hasyim, Yutaro Yamamoto, Tetsushi Mizuno, Yuna Sato, Sani Hadiyan Rasyid, Mamoru Niikura, Yu-ichi Abe, Mitsuhiro Iyori, Hiroaki Mizukami, Hisatoshi Shida, Shigeto Yoshida

**Affiliations:** 1Laboratory of Vaccinology and Applied Immunology, School of Pharmacy, Kanazawa University, Kanazawa 920-1192, Japan; kartikahardianti@gmail.com (K.H.Z.); ammarhasyim26@gmail.com (A.A.H.); yutaro59@p.kanazawa-u.ac.jp (Y.Y.); t.mizuno@staff.kanazawa-u.ac.jp (T.M.); yunasato0524@gmail.com (Y.S.); sanihadiyan92@gmail.com (S.H.R.); xxx1226827@yahoo.co.jp (Y.-i.A.); 2Department of Parasitology, Faculty of Medicine, Universitas Indonesia, Jakarta 10430, Indonesia; 3Department of Global Infectious Diseases, Graduate School of Medical Sciences, Kanazawa University, Kanazawa 920-0934, Japan; 4School of Life and Environmental Science, Azabu University, Sagamihara 252-5201, Japan; m-niikura@azabu-u.ac.jp; 5Research Institute of Pharmaceutical Sciences, Musashino University, Tokyo 202-8585, Japan; m-iyori@musashino-u.ac.jp; 6Division of Gene Therapy, Jichi Medical University, Shimotsuke 329-0498, Japan; miz@jichi.ac.jp; 7Laboratory of Primate Model, Research Center for Infectious Diseases, Institute for Frontier Life and Medical Science, Kyoto University, Kyoto 606-8507, Japan; hmyy2010@yahoo.co.jp

**Keywords:** malaria, vaccine, RTS,S, LC16m8∆, AAV, efficacy

## Abstract

**Background/Objectives**: We developed a multistage *Plasmodium falciparum* vaccine using a heterologous prime-boost immunization strategy. This involved priming with a highly attenuated, replication-competent vaccinia virus strain LC16m8Δ (m8Δ) and boosting with adeno-associated virus type 1 (AAV1). This approach demonstrated 100% efficacy in both protection and transmission-blocking in a murine model. In this study, we compared our LC16m8∆/AAV1 vaccine, which harbors a gene encoding Pfs25-PfCSP fusion protein, to RTS,S/AS01 (RTS,S) in terms of immune responses, protective efficacy, and transmission-blocking activity (TBA) in murine models. **Methods:** Mice were immunized following prime-boost vaccine regimens m8∆/AAV1 or RTS,S and challenged with transgenic *Plasmodium berghei* parasites. Immune responses were assessed via ELISA, and TB efficacy was evaluated using direct feeding assays. **Results**: m8∆/AAV1 provided complete protection (100%) in BALB/c mice and moderate (40%) protection in C57BL/6 mice, similar to RTS,S. Unlike RTS,S’s narrow focus (repeat region), m8∆/AAV1 triggered antibodies for all PfCSP regions (N-terminus, repeat, and C-terminus) with balanced Th1/Th2 ratios. Regarding transmission blockade, serum from m8∆/AAV1-vaccinated BALB/c mice achieved substantial transmission-reducing activity (TRA = 83.02%) and TB activity (TBA = 38.98%)—attributes not observed with RTS,S. Furthermore, m8∆/AAV1 demonstrated durable TB efficacy (94.31% TRA and 63.79% TBA) 100 days post-immunization. **Conclusions**: These results highlight m8∆/AAV1′s dual action in preventing sporozoite invasion and onward transmission, a significant advantage over RTS,S. Consequently, m8∆/AAV1 represents an alternative and a promising vaccine candidate that can enhance malaria control and elimination strategies.

## 1. Introduction

Malaria, caused by *Plasmodium* parasites, continues to pose a significant global health challenge, especially in endemic areas such as sub-Saharan Africa, Southeast Asia, and Latin America [1]. Despite the notable progress in malaria control efforts, the disease continues to exert a heavy toll on human populations, with an estimated 249 million cases and 608,000 malaria-related deaths reported in the World Malaria Report 2023 [2]. As there are few effective vaccines, these numbers highlight the urgent need for innovative preventive strategies to combat malaria and reduce its devastating impact on vulnerable communities [3].

RTS,S/AS01 (GSK, Rixensart, Belgium), referred to throughout as RTS,S, the first malaria vaccine to be recommended by the WHO, targets the circumsporozoite protein (CSP), a key protein expressed on the surface of sporozoites [4]. RTS,S primarily aims to stop the malaria parasite from reaching the liver and eventually entering the bloodstream [5], and its optimal immunogenicity is observed with the potent adjuvant AS01 [6]. Phase 3 clinical data indicated its modest levels of efficacy in a target population [7]. In October 2023, WHO recommended a second malaria vaccine, R21 nanoparticle in the saponin adjuvant Matrix-M^TM,^ RTS [8]. Unlike RTS,S, which contains a higher proportion of HBsAg relative to the fusion protein, R21 exclusively comprises the CSP-HBsAg fusion protein. This design allows R21 to present a greater density of CSP on its surface, a characteristic that might be responsible for a more robust generation of CSP-specific antibodies [9]. However, it is essential to note that R21 lacks monophosphoryl lipid A (MPL), which has been shown to improve memory T-cell responses. The absence of MPL in R21 may impact the longevity and quality of the immune response, potentially affecting the vaccine’s efficacy in inducing long-term protection [10].

The parasite load is considerably lower during the pre-erythrocytic phase than the blood stage of infection [11]. The scarcity of parasites at this stage makes them an optimal target for intervention and eradication within the parasite’s life cycle [12]. Nevertheless, the potential for a single evasive sporozoite to initiate a blood-stage infection renders vaccines targeting the sporozoite stage somewhat imperfect [13]. Consequently, even the most robust immune response against a singular antigen might not guarantee absolute sterile immunity in humans [14]. Therefore, adopting a vaccination approach that incorporates multiple antigens to target a multistage lifecycle may be necessary to attain a high level of protection [15,16].

The significance of transmission-blocking vaccines (TBVs) is undeniable [17]. TBVs obstruct the malaria parasite’s ability to complete its developmental cycle within the mosquito, effectively stopping its spread from one person to another [18]. By blocking transmission, these vaccines have the potential to significantly reduce, if not eradicate, malaria in endemic areas [19]. Modeling data suggest that high levels of coverage across all age groups are likely to be needed to achieve a measurable impact on transmission. Focusing solely on vaccinating children may not be sufficient to realize the full potential of TBVs in reducing malaria transmission. Furthermore, a multistage malaria vaccine that targets the various stages of the parasite’s life cycle offers a promising approach [16,20]. Such vaccines can provide broader protection by preventing the disease in vaccinated individuals and reducing the chance of parasite transmission to others [21]. This dual personal protection and transmission-blocking (TB) approach can accelerate efforts toward malaria elimination [22].

Viral-vectored vaccines provide significant benefits and stand out as promising options for tackling pathogens that have challenged conventional protein-in-adjuvant strategies such as RTS,S and the R21 vaccine [23]. They hold great promise for surmounting the challenges faced in eradicating malaria [12]. Importantly, these vaccines can elicit cytotoxic CD8^+^ T-cell responses, crucial for neutralizing intracellular parasites, most notably *Plasmodium* species, thus positioning them with a distinct advantage compared to protein-in-adjuvant vaccines [12,24]. We previously developed a multistage malaria vaccine based on a replication-competent vaccinia virus, LC16m8∆ (m8∆), and an adeno-associated virus type I (AAV1) that targeted the pre-erythrocytic stage PfCSP as well as the sexual stage Pfs25 [25]. Our study demonstrated that the m8∆/AAV1-Pf(s25-CSP) multistage malaria vaccine conferred sterile protection against sporozoites and high levels of TB activity (TBA) [26].

This study compared the immunogenicity, protective efficacy, and TBA of the m8∆/AAV1 vaccine in two murine models with those of the established malaria vaccine RTS,S. We aimed to clarify the potential of the m8∆/AAV1 vaccine as a novel tool in the battle against malaria and assess how it measures up against the current benchmark malaria vaccine, RTS,S. This research could contribute significantly to our understanding of novel malaria vaccine candidates and ultimately shape the future of malaria prevention strategies.

## 2. Materials and Methods

### 2.1. Parasites and Animals

For the challenge study, we employed transgenic *Plasmodium berghei* parasites that expressed PfCSP (designated PfCSP-Tc/Pb) [27,28]. To conduct the TB assays, we utilized double-transgenic *P. berghei* parasites that carried the Pfs25-PfCSP fusion gene (designated Pfs25-PfCSP/Pb).

To create the transfer plasmid for developing the Pfs25-PfCSP/Pb transgenic parasite line by replacing the native PbCSP with Pfs25DR3 [29], we inserted a 0.6-kb synthetic puromycin N-acetyltransferase sequence (GenBank: AY438700) with a partial *P. berghei* chromosome sequence into the *Spe*I/*Bam*HI site of pBS-5′UTR-PfCSP-DHFR-3′UTR to generate pBS-5′UTR-PfCSP-sPuro [30]. To add the T-cell epitope, a 1.5-kb fragment was excised from pBS-5′UTR-PfCSP-T-cell by digestion with *Xma*I/*Sfu*I and cloned into the *Xma*I/*Sfu*I site of pBS-5′UTR-PfCSP-sPuro to generate pBS-5′UTR-PfCSP-T-cell-sPuro [31].

The transgenic Pfs25DR3_PfCSP/Pb parasite was generated by transfection of Pfs25DR3/Pb with a linearized pBS-5′UTR-PfCSP-Puro-3′ plasmid, as described previously [31]. For genotype analysis, the genomic DNA of Pfs25DR3_PfCSP/Pb was extracted, and replacement of the *pbcsp* gene with the *pfcsp* gene was confirmed by PCR, as described in a previous study [32]. These parasites were kept and maintained in the Laboratory of Vaccinology and Applied Immunology at Kanazawa University. To evaluate immune response, level protection, and TB efficacy, we employed 6-week-old female BALB/c or C57BL/6 mice (body weight (BW) = 15~20 g). All mice were acquired from Japan SLC (Hamamatsu, Shizuoka, Japan). To initiate parasite infection in *Anopheles stephensi* mosquitoes (SDA 500 strain), we allowed the mosquitoes to feed on mice previously infected with the transgenic parasites.

### 2.2. Vaccines

LC16m8∆-Pf(s25-CSP) and AAV1-Pf(s25-CSP) (designated m8∆ and AAV1, respectively, for this study) have been described previously [16,33]. Clinical-grade recombinant RTS,S protein and the AS01 adjuvant system were provided by GlaxoSmithKline (GSK) Biologicals (Rixensart, Belgium) [34,35]. The RTS,S vaccines, depending on the dose, between 0.05 µL and 5 µL of lyophilized RTS,S protein 1 mg/mL stock were mixed with AS01 adjuvant system for a 50 µL total injection volume and incubated for 30 min at 4 °C, per manufacturer’s instructions.

### 2.3. Immunization

Six-week-old female C57BL/6 or BALB/c mice were housed in the Animal Facility of Kanazawa University for at least 1 week before receiving vaccinations. For dose optimization, three groups of BALB/c mice (*n* = 10/group) were immunized with three doses of RTS,S via intramuscular (IM) injection into the right posterior limb at a dose of 5 µg, 1.6 µg, or 0.5 µg, respectively, at 4-week intervals [9,36]. PBS alone was used for priming, first time boosting, and second time boosting in the control group (Table 1).

For a head-to-head comparative study of the m8∆/AAV1 vaccine versus the RTS,S vaccine, BALB/c mice (*n* = 10/group) or C57BL/6 mice (*n* = 10/group) were primed with m8∆ through tail scarification using a bifurcated needle and a dose of 1 × 10^7^ plaque-forming units (PFU) per mouse (Table 2). After 6 weeks (42 days), the mice were received an IM booster dose of AAV1 vaccine into the right hindlimb, at 1 × 10^10^ vg/mouse, as described in the previous study [25]. For RTS,S vaccination, three shots of a low dose of 0.5 µg RTS,S was IM administered into the right hind limb at 4-week intervals. The control group was immunized with three shots PBS alone following RTS,S schedule (Table 2).

### 2.4. Enzyme-Linked Immunosorbent Assay

The total IgG titers for both PfCSP and Pfs25 were measured using enzyme-linked immunosorbent assay (ELISA) following established protocols [28,33]. The full-length PfCSP and truncated-PfCSP (N-terminal, repeat, and C-terminal) proteins were purified using an *Escherichia coli* expression system, as described previously [37]. The Pfs25 protein was generated using a wheat germ cell-free protein expression system (Cell Free Sciences, Matsuyama, Japan) as described previously [37,38]. Serum samples were obtained from tail vein blood collected from immunized mice 1 day prior to the first boost of RTS,S or 1 day before the boost of AAV1 and challenge phases.

For ELISA, 96-well plates (Costar EIA/RIA polystyrene plates, Corning Inc., Corning, NY, USA) were coated with 0.4 µg/well of recombinant PfCSP or 0.8 µg/well of recombinant Pfs25 and left overnight at 4 °C. Following this, the plates were blocked using 1% bovine serum albumin (BSA) in PBS for 1 h before being inoculated with serum samples. Negative and positive control monoclonal antibodies (mAb 2A10 and 4B7) were serially diluted in PBS containing 1% BSA and then incubated on the plates for 1 h at room temperature. After three washes with PBST and one wash with PBS, the plates underwent incubation with a secondary antibody, horseradish peroxidase-conjugated anti-mouse IgG (Bio-Rad) or IgG subclass (IgG1, IgG2a, IgG2b, and IgG2c, Southern-Biotech, Birmingham, AL, USA), diluted to 1:2000 in PBS containing 1% BSA at room temperature for 1 h.

Subsequently, the plates were developed using a peroxidase substrate solution composed of H_2_O_2_ and 2,2′-azino-bis(3-ethylbenzothiazoline-6-sulfonate). The endpoint titers were determined as the reciprocal of the maximum dilution for which the optical density at 414 nm was equal to 0.15 U above the negative control value (<0.1). All mice used in this study were seronegative before immunization.

### 2.5. Parasite Challenge Test

Four weeks following the final immunization, the mice were intravenously (IV) challenged with 1 × 10^3^ or 1 × 10^2^ PfCSP-Tc/Pb or 1 × 10^4^ Pfs25-PfCSP/Pb sporozoites suspended in RPMI 1640 medium (Gibco Life Technologies, Tokyo, Japan). The preparation of sporozoites followed previously established protocols [39]. Infection progression was observed from day 4 to day 14 by applying Giemsa staining to thin blood smears obtained from the mice’s tails. A minimum of 20 fields (at a magnification of ×1000) were examined before designating a mouse as malaria-negative. Protection was defined as the absence of blood-stage parasitemia on day 14 post-challenge. The duration necessary to achieve 1% parasitemia was determined as described elsewhere [40].

### 2.6. Transgenic Sporozoite Neutralization Assay

A transgenic sporozoite neutralization assay (TSNA) was conducted in a total volume of 50 µL comprising 1 × 10^4^ PfCSP-Tc/Pb sporozoites in 10 µL of dissection medium, along with 10 µL of either pre-immune or immune sera from BALB/c mice and 30 µL of RPMI (-). Simultaneously, controls were set up using an equivalent number of sporozoites that were incubated either in a complete medium or in the presence of 10 µg/mL of mAb 2A10, specific for the repeats of *P. falciparum*. All neutralization experiments involved a 40-min incubation of sporozoite mixtures on ice and were duplicated. The sporozoites were introduced to HepG2 cell cultures, which were maintained at 37 °C with 5% CO_2_ for 3 days, with the culture medium was refreshed every 24 h.

After 72 h, the cells underwent treatment with trypsin-EDTA solution and two washes with sterile RNase-free PBS. Total RNA extraction was executed using a mini-RNA extraction kit (RNeasy Plus Mini kit, Qiagen, Hilden, Germany) as per the manufacturer’s guidelines. RNA elution was performed using nuclease-free water (50 µL volume), and total RNA was quantified. For each sample, 0.5 µg of total RNA underwent reverse transcription using an RT-PCR kit (GoTaq 1-Step RT-qPCR System, Promega, Madison, WI, USA) with random hexamers. A portion of the resulting cDNA was utilized for real-time RT-PCR amplification of the *P. berghei* 18S rRNA gene and the human *GAPDH* gene. The amplification of the 18S rRNA gene employed the primers forward 5′-CCAACACTTAGTCGGCATAGTT-3′ and reverse 5′-GGAGACAAACAACTGCGAAA-3′. The primers utilized to amplify the *GAPDH* gene were forward 5′-TGCCCCCATGTTCGTCATG-3′ and reverse 5′-TGTGGTCATGAGTCCTTCC-3′. The reaction’s temperature profile was 37 °C for 15 min, followed by 95 °C for 10 min, and then 40 cycles of denaturation at 95 °C for 10 s, and annealing/extension at 65 °C for 30 s.

The precise quantification of parasite-derived 18S cDNA molecules in this assay was established via linear regression analysis using the Ct values derived from infected HepG2 culture samples and those from a standard curve developed with known quantities of plasmid 18S cDNA. Normalization of the various samples for real-time PCR (RT-PCR) was achieved utilizing the *GAPDH* gene, which was co-amplified as an internal control along with the parasite 18S rRNA.

### 2.7. In Vivo Bioluminescence Imaging System

BALB/c mice were immunized with m8∆/AAV1 at 6-week intervals or with three doses of RTS,S at 4-week intervals. The control group was injected with three shots of PBS IM following RTS,S schedule. Four weeks following the final immunization, mice were IV challenged with 3 × 10^4^ PfCSP-Tc/Pb-Luciferase sporozoites. The Lumina LT in vivo bioluminescence imaging system (IVIS) (PerkinElmer, Waltham, MA, USA) was employed at 24 h, 48 h, 72 h, 96 h, and 120 h post-infection to monitor luciferase expression in the entire body following established procedures [33,41].

### 2.8. Histopathology and Immunohistochemistry

We performed immunohistochemistry to examine the ability of m8∆/AAV1 or RTS,S to prevent cerebral malaria. On day 8, brains from each group of mice were removed following IV challenge with 1 × 10^3^ Pfs25-PfCSP/Pb sporozoites and then fixed in 4% paraformaldehyde for 24 h and embedded in paraffin. Serial 4-μm-thick horizontal sections were made. To detect intracellular adhesion molecule 1 (ICAM-1) along the endothelial lining, immunohistochemical staining was performed by incubating the sections overnight at 4 °C with specific rabbit polyclonal antibodies against ICAM-1 (Abcam, Cambridge, MA, USA) diluted to 1:2000. After being washed with TBS 0.025% Triton X-100, the sections were incubated with the secondary antibody, horseradish peroxidase-conjugated goat-anti-rabbit (H + L) 1:1000, at room temperature for 1 h. Finally, the sections were washed and mounted. ICAM-1-positive vessels were visualized by microscopy at ×400 magnification. The number of positive vessels in 20 fields was counted for each mouse.

### 2.9. TB Assay

TBA was evaluated using direct feeding assays (DFAs) as described previously [28]. BALB/c or C57BL/6 mice were immunized with m8∆/AAV1 or RTS,S. Four weeks after the final immunization, BALB/c and C57BL/6 mice were IV challenged with 1 × 10^3^ and 1 × 10^2^ PfCSP-Tc/Pb sporozoites, respectively (Tabel 2). The sera from protected mice were collected and pooled. The 500 μL pooled sera of BALB/c or C57BL/6 mice were injected into naïve C57BL/6 mice (*n* = 3/group). After 1 h, the mice were IV challenged with double-transgenic Pfs25-PfCSP/Pb sporozoites. When parasitemia levels reached 5–10%, 50–60 starved *An. stephensi* mosquitoes were allowed to feed on each infected mouse.

We then individually examined the TB efficacy of the m8∆/AAV1 vaccine against double-transgenic Pfs25-PfCSP/Pb parasites [26,28]. BALB/c mice were immunized with m8∆/AAV1 vaccine. Four weeks after the final immunization, BALB/c mice were IV challenged with 1 × 10^4^ Pfs25-PfCSP/Pb parasites. Then, the protected mice (*n* = 5) were kept until 13-weeks after the final immunization. Mice were re-challenged with 2 × 10^6^ double-transgenic Pfs25-PfCSP/Pb infected red blood cells (iRBCs) intraperitoneal (i.p.). On day 3, 20–30 starved *An. stephensi* mosquitoes were allowed to feed on each infected mouse.

For both experiments described above, unfed mosquitoes were removed 5–6 h post-feeding. The mosquitoes were then maintained on a fructose solution [8% (*w*/*v*) fructose, 0.05% (*w*/*v*) p-aminobenzoic acid] at 19–22 °C and 50–80% relative humidity. On days 11–12 post-feeding, the mosquito midguts were dissected, and oocyst prevalence and intensity were recorded. The number of oocysts was counted for each mouse, and the mean oocyst intensity was calculated. As described previously, these numbers were compared for inhibition calculations with mice without immunization [28]. The percentage inhibition of mean oocyst intensity (transmission-reducing activity; TRA) was calculated as follows: 100 × [1 − (mean number of oocysts in the test group/mean number of oocysts in the control group)]. In addition, the percentage oocyst prevalence inhibition (transmission-blocking activity; TBA) was evaluated as 100 × [1 − (proportion of mosquitoes with any oocysts in the test group)/(proportion of mosquitoes with any oocyst in the control group)] [42].

### 2.10. Statistical Analysis

Statistical analysis was performed using GraphPad Prism version 8.0 for Mac OS. An unpaired *t*-test was employed to assess the differences between the two groups. All ELISA end-point titers were log_10_-transformed before analysis. Antibody durability was evaluated using Sidak’s multiple comparison test. The proportion of mice not reaching 1% parasitemia was analyzed using a Kaplan–Meier log-rank (Mantel–Cox) test. TRA differences between groups were analyzed using a Mann–Whitney *U*-test. TBA differences between groups were assessed using Fisher’s exact test.

## 3. Results

### 3.1. Schematic Representation of m8∆/AAV1 and RTS,S Vaccines

The m8∆ or AAV1 vaccine construct consists of a fusion gene, *pfs25-pfcsp*, linked by a hinge peptide (H), Gly6Ser, between the *pfcsp* and *pfs25* genes as described previously [26]. In the m8∆/AAV1 vaccine, the PfCSP comprises the full-length circumsporozoite protein, including the N-terminal, NANP-repeat, and C-terminal regions (Figure 1A). The viral-vectored vaccine construction of m8∆ and AAV1 has been described in previous studies [16,25]. This design allows the m8∆/AAV1 vaccine to elicit immune responses targeting both the pre-erythrocytic (PfCSP) and sexual (Pfs25) stages.

In contrast, the RTS,S vaccine construct includes the last 16 NANP repeats (R) and the entire C-terminal region of PfCSP, with hepatitis B surface antigen (HBsAg) particles fused to the PfCSP segment [43]. Additional HBsAg is included to enhance the vaccine construct’s immunogenicity (Figure 1B). The PfCSP segment in RTS,S also contains three T cell epitopes (T), a highly variable CD4+ and CD8+, and a conserved CD4+ T cell epitopes at the C-terminus. These epitopes are critical for the vaccine’s ability to induce both humoral and cellular immune responses to prevent the malaria parasite’s liver-stage infection [44,45].

### 3.2. Low-Dose RTS,S Elicits Robust Humoral Immune Response and Sterile Protection

To evaluate the immunogenicity of low-dose RTS,S vaccination, BALB/c mice were administered different amounts of RTS,S protein (5 µg, 1.6 µg, or 0.5 µg) along with the AS01 adjuvant. This three-dose regimen, spaced 4 weeks apart, was followed by a challenge with 1 × 10^3^ transgenic *P. berghei* sporozoites expressing PfCSP (PfCSP-Tc/Pb). The levels of PfCSP-specific antibodies were evaluated in a day before the first boost and challenge. As a control, PBS control mice were similarly subjected to the challenge. The results indicated that the 0.5 µg dose of RTS,S triggered robust anti-PfCSP total IgG titers at 4 weeks after the last dose (ranging from 2.3 × 10^5^ to 2.5 × 10^6^, mean = 1.1 × 10^6^) that were comparable to the antibody levels induced by higher doses of 1.6 µg (ranging from 1.2 × 10^5^ to 1.6 × 10^6^, mean = 7.1 × 10^5^) or 5.0 µg (ranging from 3.3 × 10^5^ to 1.5 × 10^6^, mean = 6.2 × 10^5^). No significant difference was observed, and there was no discernible elevation in anti-PfCSP total IgG titers with the increase in vaccine dose (Figure 2A).

Following sporozoite challenge, all vaccine doses led to a notable delay in reaching 1% parasitemia compared to PBS control mice. Additionally, each dosage group exhibited 90% protection (Figure 2B). This compelling outcome prompted the selection of the 0.5 µg dose of RTS,S for a subsequent comparison of vaccine efficacy with our viral-vectored vaccine platform, m8∆/AAV1.

### 3.3. Long-Lasting Humoral Responses and Sterile Protection

To assess the protective efficacy of m8∆/AAV1 and RTS,S vaccines, immunized mice were IV challenged with 1 × 10^3^ PfCSP-Tc/Pb sporozoites 4 weeks after the last immunization. Following the challenge, all 10 mice (100%, *p* < 0.0001) were protected in both vaccine groups, m8∆/AAV1 and RTS,S (Figure 3A). These results indicate that the m8∆/AAV1 vaccine effectively blocked sporozoite invasion of the liver cells and prevented the establishment of blood-stage infections as strongly as the RTS,S vaccine. This finding underscores the potential of both vaccines to disrupt the parasite’s life cycle, preventing the infection progressing to the symptomatic phase.

Next, we assessed the antibody durability of the protected mice. Sera were collected at different time points from both vaccination groups. Both m8∆/AAV1 and RTS,S induced high total anti-PfCSP IgG titers (2.9 × 10^5^–6.9 × 10^5^ and 1.1 × 10^6^–2.2 × 10^6^, respectively) that were sustained until 17 weeks after the last immunization. There was no significant difference in the levels of antibody durability between the m8∆/AAV1 and RTS,S vaccines (Figure 3B). These results indicate that the m8∆-prime/AAV1-boost vaccination regimen successfully induced robust and durable anti-PfCSP humoral immune responses that were comparable with those of RTS,S.

### 3.4. A Broad Humoral Immune Response across Truncated PfCSP

To evaluate the proportion of humoral immune responses to truncated PfCSP proteins, we examined the anti-truncated-PfCSP IgG titers raised to recombinant truncated PfCSP proteins produced by the *E. coli* system (N-terminus (N), repeat region (R), and C-terminus I protein) (Figure 4A). The results demonstrated that the m8∆/AAV1 vaccine induced a more evenly distributed IgG response to truncated-PfCSP antigens (anti-C = 51.54%, anti-R = 39.30%, and anti-N = 9.16%) (Figure 4B). In contrast, the RTS,S vaccine elicited a predominantly repeat-region-focused response (anti-R = 68.15%, anti-C = 31.74%) and a negligible response to the N-terminal region (anti-N = 0.11%) (Figure 4C).

These data suggest that the m8∆/AAV1 vaccine induced a broader immune response across the different PfCSP regions, which could potentially induce more effective protection. While the RTS,S vaccine, which is more focused on the repeat region, may provide a more intense targeted response. However, it is important to note that a broader immune response might also result in a less concentrated response to the key protective epitopes in the repeat region, potentially leading to less overall protection.

### 3.5. IgG Subclass Distribution with m8∆/AAV1 Vaccine

We addressed the IgG subclass distribution elicited by the anti-PfCSP repeat responses of the m8∆/AAV1 and RTS,S vaccine groups. This aspect is informative, as the subclass of IgG antibodies can influence the type of immune response, i.e., T-helper 1 (Th1) or T-helper 2 (Th2), which is critical for the effectiveness of a vaccine.

In BALB/c mice, the m8∆/AAV1 vaccine led to a relatively even distribution of IgG subclasses (IgG2a = 50.78%, IgG1 = 27.32%, and IgG2b = 21.90%), suggesting a balanced Th1/Th2 response occurred (Figure 5A,B). In stark contrast, the RTS,S vaccine group exhibited a strong bias towards a Th2 response, with a high IgG1 dominance (IgG1 = 94.49%, IgG2b = 5.24%, and IgG2a = 0.27%) (Figure 5A,C).

In C57BL/6 mice, which are more Th1-prone, the m8∆/AAV1 vaccine maintained a relatively balanced IgG subclass response with IgG2b (42.23%), IgG2c (29.98%), and IgG1 (27.79%), indicating a Th1/Th2 mixed response (Figure 5D,E). Conversely, the RTS,S vaccine in C57BL/6 mice induced a highly Th1-skewed response, with IgG2b (80.14%), IgG2c (18.27%), and very low IgG1 (1.59%) (Figure 5D,F).

These findings suggest that the m8∆/AAV1 vaccine may induce a more versatile and balanced immune response by engaging both Th1 and Th2 pathways. This contrasts with the RTS,S vaccine, which promoted a strong Th2 bias in BALB/c mice and a dominant Th1-skewed response in C57BL/6 mice.

### 3.6. Vaccine-Induced Immune Sera Neutralize Sporozoite Invasion of Liver Cells

To show the neutralizing ability of anti-PfCSP antibodies in the sera from immunized mice inoculated with the m8∆/AAV1 or RTS,S vaccine, we performed an in vitro TSNA using HepG2 cells. We incubated sporozoites with naïve mouse sera or with 50- or 5-times diluted immunized mouse sera. The results showed that the sera from mice immunized with the m8∆/AAV1 or RTS,S vaccine significantly reduced the relative percentage of 18S rRNA copy number (Appendix A).

Then, we evaluated the neutralizing activity of anti-PfCSP antibody induced by the m8∆/AAV1 or RTS,S vaccine by IVIS. PfCSP-Tc/Pb transgenic sporozoite expressing luciferase (PfCSP-Tc/Pb-Luc) was IV administered into the tail vein (3 × 10^4^ sporozoite/mouse) of BALB/c mice (*n* = 2/group) on day 0. Groups of mice immunized with m8∆/AAV1 or RTS,S vaccine did not show luciferase expression until 120 h after the PfCSP-Tc/Pb-Luc challenge (Appendix A). In contrast, in the naïve mice, luciferase expression started at 24 h (mean: 3.43 × 10^4^ photons/s/cm^2^/sr) and was distributed throughout the whole body by 120 h (mean: 1.07 × 10^7^ photons/s/cm^2^/sr) after PfCSP-Tc/Pb-Luc challenge (Appendix A). These results indicate that the anti-PfCSP induced by m8∆/AAV1 significantly inhibited sporozoite invasion of the liver cells and was as effective as the RTS,S vaccine. These results suggest that the immune responses elicited by the m8∆/AAV1 vaccine can target critical stages of infection, further reinforcing the evidence for its efficacy in preventing malaria.

### 3.7. Vaccines Provide Moderate Protection in C57BL/6 Mice

We performed a comparative study of two-dose m8∆/AAV1 and three-dose RTS,S vaccinations in C57BL/6 mice, which is a more sensitive model than BALB/c mice [36]. Both vaccines induced high anti-PfCSP IgG titers (4.7 × 10^5^ and 5.2 × 10^5^ after the last immunization, respectively) (Figure 6A). However, 4 mice (40%, *p* < 0.0001) in each vaccine group were protected against IV challenge with 100 PfCSP-Tc/Pb sporozoites (Figure 6B). These data indicate that the m8∆/AAV1 vaccine offered a significant protective advantage in a two-dose regimen compared to the three-dose RTS,S vaccine, suggesting the m8∆/AAV1 vaccine can provide a more efficient vaccination strategy against malaria.

### 3.8. Double-Transgenic Pfs25-PfCSP/Pb Parasite Construction

The m8∆/AAV1 vaccine was designed as a multistage vaccine effective for both protection and TB. We constructed a double-transgenic Pfs25DR3_PfCSP/Pb (designated as Pfs25-PfCSP/Pb) parasite to support our next investigation. Using a plasmid that carries both the complete *pfcsp* gene and the gene for puromycin *N-*acetyltransferase (which grants puromycin resistance), the Pfs25-PfCSP/Pb parasite strain was created (as shown in Figure 7A). The replacement cassette from *P. falciparum* featured a CSP sequence (amino acids 24 to 397) specific to the *P. falciparum* 3D7 strain (GenBank: XP_001351122). *Pb*Pfs25DR3 parasites in their blood stage were in vitro transfected using a plasmid pre-treated and linearized with a restriction enzyme following established methods [46]. The substitution of the gene and the integration of the *pfcsp* gene cassette into the genome were verified by PCR of the genetically modified Pfs25-PfCSP/Pb parasite (Figure 7B). Immunofluorescence assays indicated that the PfCSP protein was displayed on the surface of each sporozoite retrieved from the salivary glands of mosquitoes infected with Pfs25-PfCSP/Pb. Pfs25 expression was also observed on the surface of in vitro cultured gametocytes (Figure 7C). The parasite maintained full viability across its entire lifecycle.

### 3.9. M8∆/AAV1 Vaccine Matches RTS,S’s Efficacy in Preventing Cerebral Malaria

In the quest to combat cerebral malaria, this study has highlighted the efficacy of the m8∆/AAV1 and RTS,S vaccines in C57BL/6 mouse models. Immunohistochemical analysis revealed that m8∆/AAV1 or RTS,S vaccination significantly reduced the expression of ICAM-1 in the brain vasculature, a marker often associated with cerebral malaria pathology (Appendix A). This indicated a possible decrease in cerebral inflammation, a critical factor in malaria pathogenesis. A quantitative assessment showed that the vaccinated groups (m8∆/AAV1 and RTS,S) had fewer ICAM-1-positive vessels than the infected, non-vaccinated controls (Appendix A). These data suggest that both vaccines can mitigate the endothelial activation associated with severe malaria.

### 3.10. TB Activity

An intriguing observation arose when evaluating anti-Pfs25 antibody responses and TB efficacy. BALB/c or C57BL/6 mice were immunized with the m8∆/AAV1 vaccine at 6-week intervals or the RTS,S vaccine at 4-week intervals. The m8∆/AAV1 vaccine induced a high-level anti-Pfs25 antibody response in BALB/c mice (5.7 × 10^4^–4.3 × 10^5^; mean: 2.1 × 10^5^) and C57BL/6 mice (2.7 × 10^3^–2.7 × 10^5^; mean: 6.7 × 10^4^) (Figure 8A).

Next, the sera of BALB/c or C57BL/6 mice, those were protected against challenge with transgenic parasite PfCSP-Tc/Pb, were pooled and passively transferred to two groups of naïve C57BL/6 mice. After 1 h, mice were injected with double transgenic parasites Pfs25-PfCSP/Pb IV. A total of 50~60 *An. stephensi* mosquitoes were allowed to feed on the mice that received the pooled sera. The sera from BALB/c mice immunized with m8∆/AAV1 induced TRA and TBA (83.02% and 38.98%, respectively). While the sera from C57BL/6 mice immunized with m8∆/AAV1 induced moderate TRA and a low level of TBA (56.53% and 9.02%, respectively). In contrast, pooled sera from the mice immunized with RTS,S did not reduce parasite transmission to mosquitoes (Figure 8A and Table 3). These data indicate that the m8∆/AAV1 vaccine’s capability to induce high anti-Pfs25 antibody responses in BALB/c and C57BL/6 mice correlated with TBA, particularly in BALB/c mice, highlighting the potential of m8∆/AAV1 as a vaccine candidate for reducing malaria transmission.

To further our investigation, BALB/c (*n* = 5/group) mice were immunized with m8∆/AAV1. Four weeks after the last immunization, the mice were IV challenged with 1 × 10^4^ of double-transgenic Pfs25-PfCSP/Pb parasites. We expected all mice to be infected, but none were determined to be infected after 14 days of blood smear testing. Therefore, we kept the protected mice until 91 days (13 weeks) after the last immunization, when they were then infected with 2 × 10^6^ Pfs25-PfCSP/Pb iRBCs i.p. We collected sera from the mice before the mosquito-feeding challenge and examined the total anti-Pfs25 IgG titers. A decline in antibody levels was observed pre-DFA (mean: 3.51 × 10^4^) compared to those measured 4 weeks after the boost time point (mean: 8.03 × 10^4^) (Figure 8B). These sustained titers suggest that, while there was some waning, the m8∆/AAV1 vaccine maintained a durable immunogenic profile over time.

Next, the mice were subjected to feeding by ~20–30 *An. stephensi* mosquitoes. The corresponding TB assay revealed a profound reduction in oocyst numbers in the mosquitoes fed on m8∆/AAV1-immunized mice, with potent TRA of 94.31% and TBA of 63.79%, indicating the vaccine’s significant long-term potential to hinder malaria transmission (Figure 8C). These findings suggest that the m8∆/AAV1 vaccine provided immediate protection against transgenic parasites and ensured long-term transmission-reducing and blocking effects, highlighting its potential for long-lasting malaria control.

## 4. Discussion

The present study was designed to evaluate the efficacy of the m8∆/AAV1 vaccine in comparison to the RTS,S vaccine in murine models to contribute to the data on its potential as a preventive tool against malaria. The m8∆/AAV1 vaccine is noteworthy for its multistage approach, providing high-level protection with a two-dose regimen. This efficiency in inducing robust immune responses highlights its promise as a versatile and potent option for malaria vaccination, emphasizing its potential to optimize immunogenicity without the need for multiple booster doses.

The sterile protection afforded by m8∆/AAV1 and RTS,S vaccines against PfCSP-Tc/Pb sporozoite challenge is congruent with the expected ability of proficient malaria vaccines to avert blood-stage infections and thus play a vital role in interrupting the *Plasmodium* parasite’s lifecycle [47]. Both vaccines demonstrated the capacity to block sporozoite invasion of the liver cells, with immune sera indicating the production of neutralizing antibodies targeting early-stage infection [48,49]. Unlike RTS,S, which primarily produces antibodies to the PfCSP repeat region, the m8∆/AAV1 vaccine induced a humoral response across all PfCSP regions [50,51]. This comprehensive response, particularly the production of anti-N-terminal antibodies, could block the hepatocyte invasion steps integral to the parasite’s life cycle [52].

The m8∆/AAV1 vaccine’s balanced IgG subclass distribution suggests it induces a versatile Th1/Th2 immune response in both Th2-prone (BALB/c) or Th1-prone (C57BL/6) mice, offering potentially broader protection against malaria [53,54]. The significant IgG2a presence is indicative of a potent intracellular pathogen defense [55]. This contrasts with the RTS,S vaccine’s Th2-biased response in BALB/c mice that is marked by a high IgG1 concentration, while inducing a strong Th1-skewed response in C57BL/6 mice, dominated by IgG2b [56]. The ability of the m8∆/AAV1 vaccine to stimulate both Th1 and Th2 pathways could yield more persistent immune memory and sustained protection and provide a nuanced immune strategy necessary for effective immunity [57].

Furthermore, the robust humoral immune responses observed in C57BL/6 mice, with their immunogenicity-specific genetic backgrounds [58], in both vaccine groups indicated the broad application potential of the vaccines [59]. The moderate protection provided by both vaccines in this mouse strain, alongside the impact of the vaccines on cerebral endothelial activation, essential for mitigating cerebral, suggests there were complex protective immune mechanisms at play [60]. C57BL/6 mice are Th1-prone, which means they predominantly generate a cell-mediated immune response driven by CD8+ T cells and the production of IFN-*γ*, making them more responsive to vaccines that rely on cellular immunity [61,62]. However, this immune polarization may limit their ability to mount an equally robust humoral (antibody) response, which is crucial for vaccine-targeting the sporozoite stage of malaria, such as RTS,S, and m8∆/AAV1 [36,63]. In contrast, BALB/c mice are Th2-prone and produce strong antibody responses, essential for neutralizing the malaria sporozoites before they invade hepatocytes. This Th2-skewed response in BALB/c mice can lead to higher efficacy in vaccines that rely on neutralizing antibodies [9].

Moreover, the reduction in ICAM-1 expression in the m8∆/AAV1 and RTS,S vaccine groups correlated with decreased cerebral endothelial activation, crucial for preventing the sequestration of infected erythrocytes and subsequent vascular inflammation [64]. Notably, despite the differences in vaccine constructs, both vaccines profoundly prevented cerebral malaria. This indicates a shared protective mechanism was involved, which warrants further investigations to optimize vaccine strategies against malaria. However, it is important to note that while mouse models provide valuable insights, they do not fully replicate human cerebral malaria. Therefore, further studies of more advanced models and human clinical trials are necessary to validate these findings.

It is worth noting the recent WHO endorsement of the R21/Matrix-M vaccine, which has demonstrated significant efficacy against malaria in phase III clinical trials [10]. Developed as a subunit vaccine with an adjuvant, this vaccine is formulated to be easily manufactured, stored, and delivered [65]. Future studies comparing m8∆/AAV1 with R21/Matrix-M could provide further insights into these vaccines’ long-term protection and transmission-blocking efficacy.

In the TB assay, the m8∆/AAV1 vaccine exhibited profound TB potential, as evidenced by the durable anti-Pfs25 humoral immune response and significant TRA and TBA in immunized mice. These results highlight the vaccine’s ability to elicit a robust immune response and to curb the spread of the parasite within a population. While modeling data suggest that a TB component alone might not lead to a meaningful reduction in transmission, the m8∆/AAV1 vaccine still shows promise for use within integrated malaria control strategies. Additionally, it is important to consider the acceptability of the AAV platform in healthy young children. The AAV platform has shown a favorable safety profile in various clinical settings, which supports its potential use in pediatric populations. These factors position m8∆/AAV1 as a promising vaccine that could play a vital role in protecting individuals and interrupting the malaria transmission cycle, contributing to the ultimate goal of malaria eradication [66].

## 5. Conclusions

Regarding protection, the two-dose m8∆/AAV1 vaccine regimen was as effective as the three-dose RTS,S vaccine in murine models and had the added advantage of inducing antibodies across all PfCSP regions and a balanced Th1/Th2 response. In C57BL/6 mice, both vaccines provided 40% protection and were less effective than in BALB/c mice. The TB efficacy of the m8∆/AAV1 vaccine may prevent the spread of mutated parasites from breakthrough infections in vaccinated individuals; therefore, m8∆/AAV1 could be a favorable choice as a multistage vaccine for malaria eradication programs.

## Figures and Tables

**Figure 1 vaccines-12-01155-f001:**
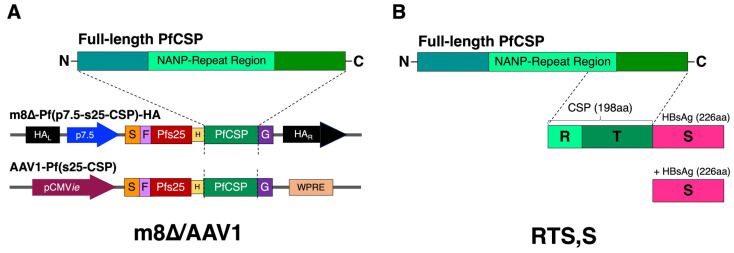
Schematic representation of the m8∆/AAV1 and the RTS,S vaccines. (**A**) The m8∆/AAV1 vaccine consists of a fusion gene, *pfs25-pfcsp*. The PfCSP segment includes the entire length of PfCSP, which contains the N-terminal, NANP-repeat, and the C-terminal region of the circumsporozoite protein (CSP). The Pfs25 targets the sexual stage of the malaria parasite for transmission-blocking (TB), while the PfCSP targets the pre-erythrocytic stage. (**B**) The RTS,S vaccine includes the last 16 NANP repeats and the entire C-terminal region of PfCSP fused to the hepatitis B surface antigen (HBsAg), with additional HBsAg particles included to enhance immunogenicity. N = N-terminal region, C = C-terminal region, R = Repeat region, T = Three T cell epitopes at C-terminal region, and S = Hepatitis B surface Antigen (HBsAg).

**Figure 2 vaccines-12-01155-f002:**
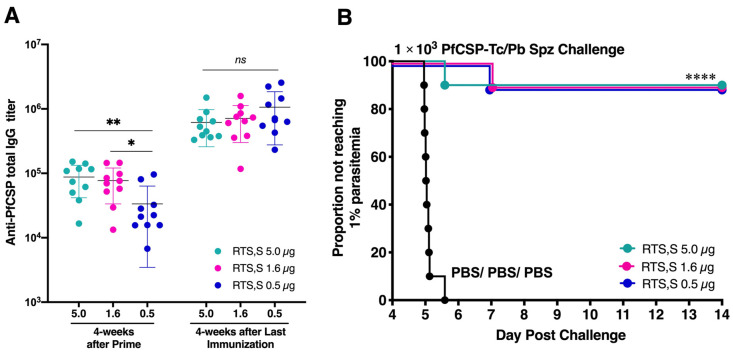
Immunogenicity and protective efficacy of RTS,S/AS01 (RTS,S) at three distinct doses. BALB/c mice (*n* = 10/group) received three immunizations, 4 weeks apart, of RTS,S at a range of doses (0.5 µg, 1.6 µg, or 5 µg). (**A**) Sera were collected 4 weeks after the prime or final immunization, and then the anti-PfCSP total IgG titer was measured by ELISA. Black lines indicate the mean, and group differences were assessed with a Mann–Whitney *U* test. * *p* < 0.05, ** *p* < 0.001, ns: not significant. (**B**) Four weeks after the final immunization, mice were challenged with 1 × 10^3^ transgenic *P. berghei* parasites expressing *P. falciparum* CSP (PfCSP-Tc/Pb). Blood-stage parasitemia was monitored from day 4 after the challenge by examining thin blood smears, and a model predicting the time to reach 1% parasitemia was generated. The absence of blood-stage parasites in the mice was confirmed on day 14 after the challenge. Results are presented in Kaplan–Meier survival curves, and *p*-values were calculated with a Kaplan–Meier log-rank (Mantel–Cox) test for each group versus the PBS (control) group. **** *p* < 0.0001.

**Figure 3 vaccines-12-01155-f003:**
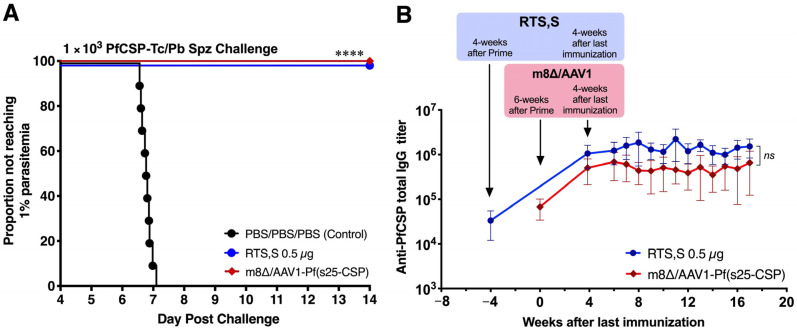
Protective efficacy and immunogenicity of m8∆/AAV1-Pf(s25-CSP) versus RTS,S/AS01 (RTS,S). BALB/c mice (*n* = 10/group) were immunized with indicated regimens at 6- or 4-week intervals (Table 2). (**A**) Four weeks after the last immunization, the mice were challenged with an IV injection of 1 × 10^3^ transgenic *P. berghei* parasites expressing *P. falciparum* CSP (PfCSP-Tc/Pb) sporozoites. Parasitemia was examined for 3 consecutive days, beginning on the fourth day post-challenge, and a model was created to predict the time required to reach 1% parasitemia. The absence of blood-stage parasites in the mice was confirmed on the fourteenth day following the challenge. Statistical analysis involved generating Kaplan–Meier survival curves, with *p*-values calculated using the Kaplan–Meier log-rank (Mantel–Cox) test for each immunized group compared to the PBS/PBS/PBS (control) group. **** *p* < 0.0001. (**B**) The durability of anti-PfCSP. BALB/c mice (*n* = 10/group) were immunized with m8∆/AAV1 or RTS,S at 6- or 4-week intervals. Individual serum samples were collected 6-week after prime of m8∆, or 4-week after prime of RTS,S, then at different time point after final immunization. Since 6-week after the last immunization, sera were collected weekly until 17-week. Anti-PfCSP total IgG titers were measured using ELISA. Each data plot was shown as the mean value of anti-PfCSP total IgG titer at different time point. The error bars were shown as a 95% confidence interval (95% CI). The discrepancies in antibody titers between the groups were assessed using Sidak’s multiple comparisons test. ns; not significant.

**Figure 4 vaccines-12-01155-f004:**
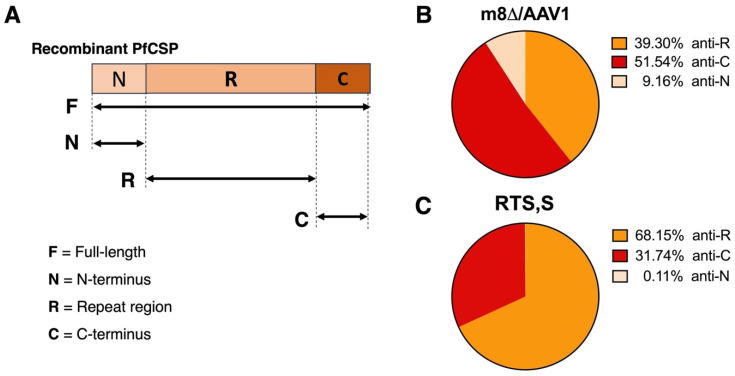
Anti-truncated PfCSP IgG titers. (**A**) Schematic representation of the recombinant truncated PfCSP N-terminus (N), repeat region (R), and C-terminus (C) proteins produced by the *E. coli* system. BALB/c mice (*n* = 10/group) were immunized with m8∆/AAV1 or RTS,S/AS01 (RTS,S) at 6- or 4-week intervals. Individual sera were collected 4 weeks after final immunization. Anti-truncated PfCSP IgG titers were measured using ELISA. (**B**) Pie chart illustrates the proportion of anti-PfCSP IgG responses to the truncated PfCSP antigens in the m8∆/AAV1 vaccine and (**C**) RTS,S vaccine groups.

**Figure 5 vaccines-12-01155-f005:**
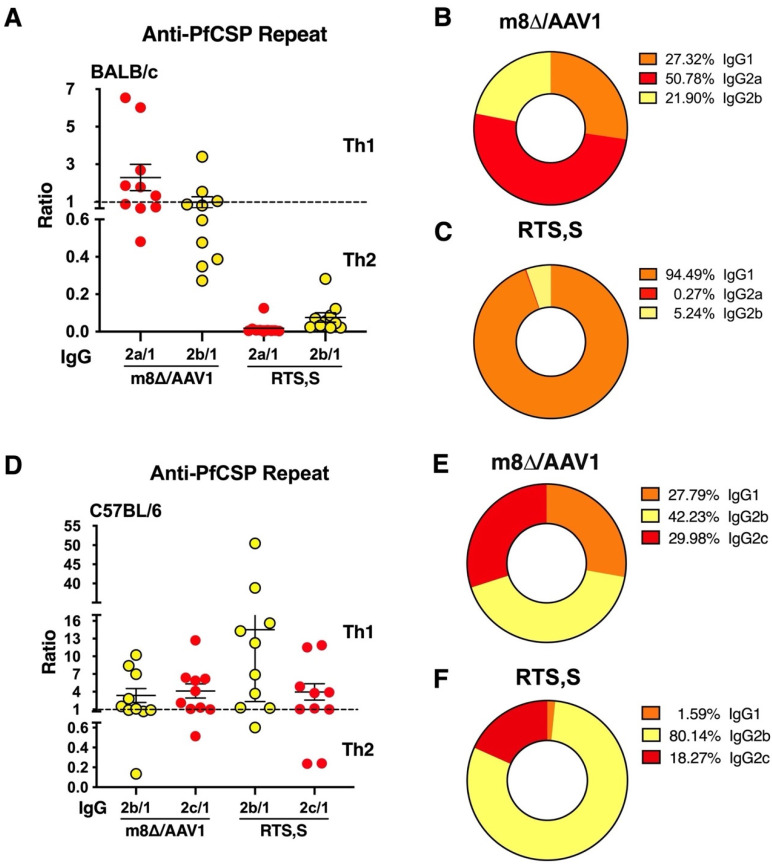
IgG subclass ratio of the anti-PfCSP repeat region. BALB/c or C57BL/6 mice (*n* = 10/group) were immunized with m8∆/AAV1 or RTS,S/AS01 (RTS,S) at 6- or 4-week intervals (Tabel 2). Individual sera were collected after final immunization. IgG subclasses against PfCSP repeats were measured using ELISA. (**A**) Ratios of IgG2a/IgG1 and IgG2b/IgG1 in BALB/c mice. (**B**) Proportional representations of IgG subclasses for m8∆/AAV1 or (**C**) RTS,S in BALB/c mice. (**D**) Ratios of IgG2b/IgG1 and IgG2c/IgG1 in C57BL/6 mice. (**E**) Proportional representations of IgG subclasses for m8∆/AAV1 or (**F**) RTS,S in C57BL/6 mice. In panel (**A**,**D**), horizontal line was shown as the mean value of IgG subclass ratio. The error bars were shown as a mean value with standard error of mean (SEM).

**Figure 6 vaccines-12-01155-f006:**
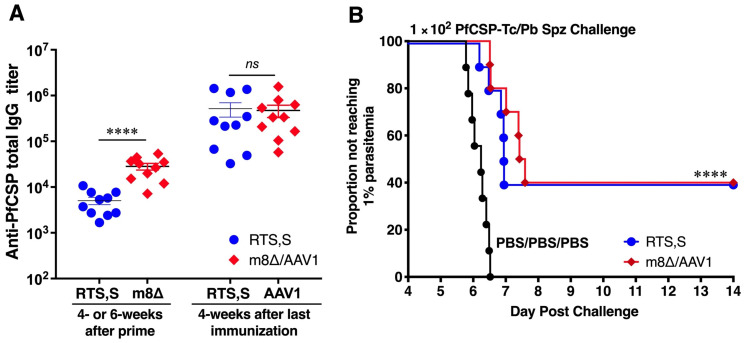
Immunogenicity and protective efficacy of m8∆/AAV1 versus RTS,S/AS01 (RTS,S). C57BL/6 mice (*n* = 10 per group) were immunized with indicated regimens at 6- or 4-week intervals (Tabel 2). (**A**) Humoral immune responses. Individual sera were collected at 4- or 6-week after the prime, then 4-week after the last immunization. Anti-PfCSP total IgG titers were measured using ELISA. (**B**) Four weeks after the last immunization, the mice were challenged with an intravenous (IV) injection of 1 × 10^2^ transgenic PfCSP-Tc/Pb sporozoites. Parasitemia was monitored for 3 consecutive days, starting from day 4 after the challenge, and a model predicting the time to reach 1% parasitemia was generated. The absence of blood-stage parasites in the mice was confirmed on day 14 after the challenge. Data in panel A are the mean ± SEM. Differences between groups were analyzed using a Mann–Whitney *U* test. **** *p <* 0.0001, ns: not significant. In panel (**B**), the statistical analysis involved creating Kaplan–Meier survival curves and calculating *p*-values using a Kaplan–Meier log-rank (Mantel–Cox) test to compare each immunized group against the PBS/PBS/PBS (control) group. **** *p* < 0.0001.

**Figure 7 vaccines-12-01155-f007:**
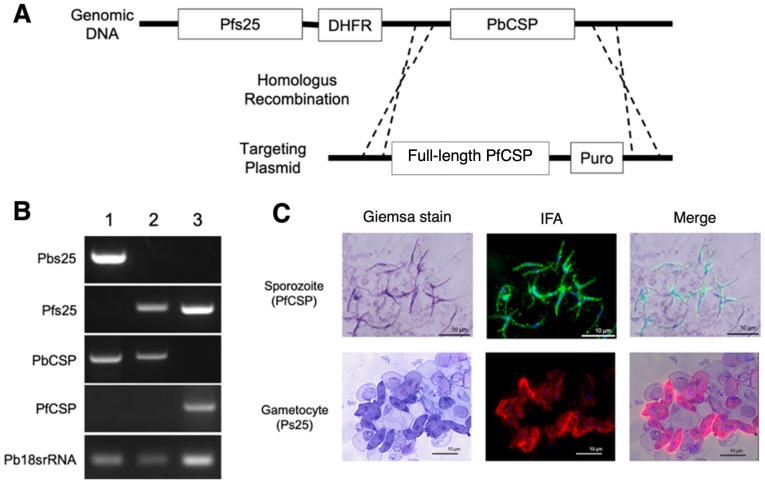
Schematic of the genomic region targeted by the PfCSP cassette. (**A**) The gene cassette consisted of the full-length PfCSP. The *pbcsp* gene was replaced with the *pfcsp* cassette and the puromycin N-acetyltransferase selectable marker. (**B**) The replacement of the wild-type *pbcsp* gene with the *pfcsp* gene in the Pfs25DR3-Pb parasite lines was confirmed through PCR using genomic DNA from WT-Pb (lane 1), Pfs25DR3-Pb (lane 2), or Pfs25DR3_PfCSP-Pb (lane 3). Pb18s rRNA was used as a loading control. (**C**) Giemsa staining and immunofluorescence assay (IFA) of Pfs25DR3_PfCSP-Pb sporozoites and gametocytes. Sporozoites were labeled with the 2A10 mAb conjugated with Alexa Flour 488 (green). Gametocytes were labeled with the 4B7 mAb conjugated with Alexa Flour 595 (red). Parasite nuclei were visualized with DAPI (blue). Scale bar = 10 μm.

**Figure 8 vaccines-12-01155-f008:**
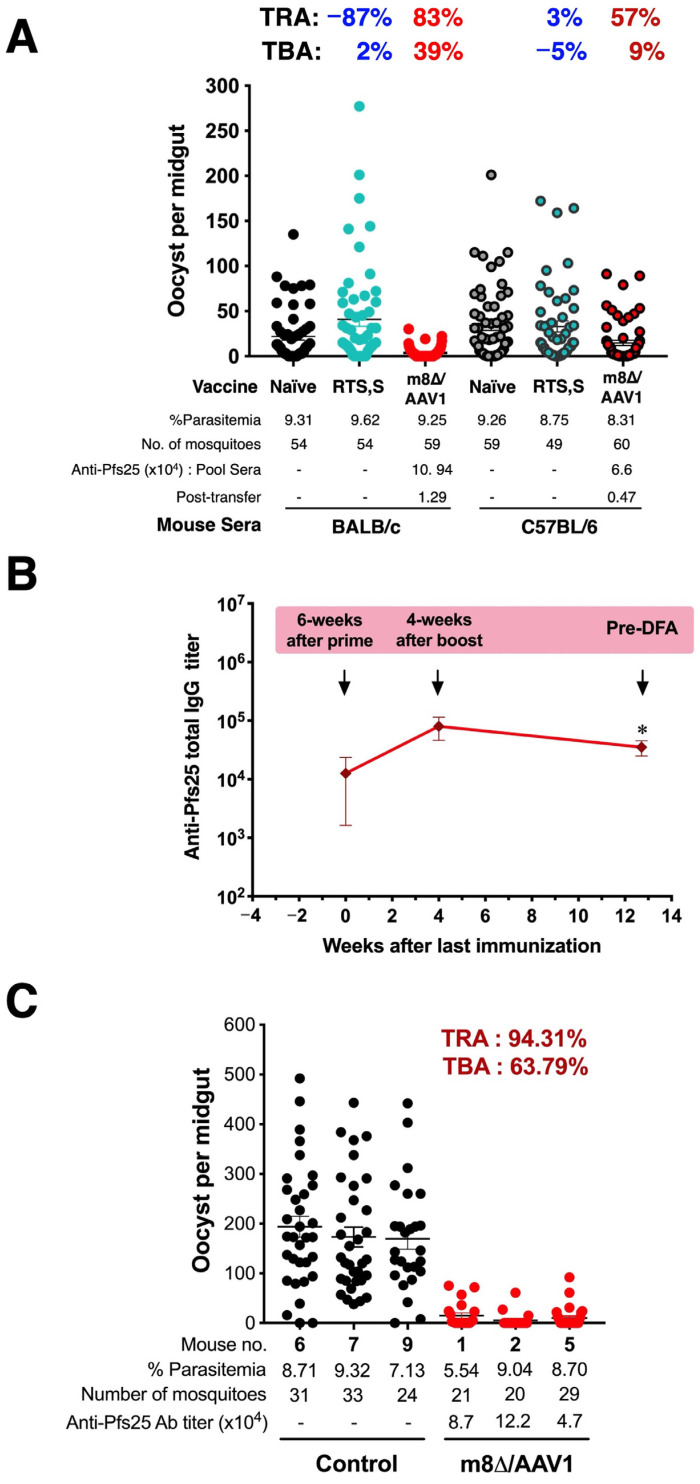
Transmission-blocking assay. (**A**) Transmission-blocking activity (TBA) of m8∆/AAV1 and RTS,S/AS01 (RTS,S). Direct feeding assays (DFAs) with BALB/c and C57BL/6 mice show differential TBA. Mice were immunized with m8∆/AAV1 or RTS,S and challenged with PfCSP-Tc/Pb sporozoites. Serum from protected mice was transferred to naïve C57BL/6 mice, which were then exposed to starved *An. stephensi* mosquitoes. (**B**) Kinetics of anti-Pfs25 antibody (Ab) titers. Sera were collected at 6 weeks after prime or 4 weeks after the last immunization, then 13 weeks after the final immunization. Differences between anti-Pfs25 at pre-DFA and 4 weeks after booster were analyzed using an unpaired *t*-test. * *p* < 0.05. (**C**) Assessment of m8∆/AAV1′s TBA using DFAs. Protected BALB/c mice challenged with double transgenic parasites, Pfs25-PfCSP/Pb, were compared to PBS control mice after exposure to starved *An. stephensi* mosquitoes. Oocyst prevalence and intensity in mosquito midguts were assessed to calculate the TRA and TBA. TRA and TBA percentages indicate the relative reduction in oocyst intensity and prevalence, respectively, compared to controls.

**Table 1 vaccines-12-01155-t001:** Immunization schedule for evaluating the efficacy of three distinct dosage levels of RTS,S/AS01 (RTS,S) in BALB/c mice.

Group	Vaccine Administration			Challenge
	Day 0 ^a^	Day 28 ^b^	Day 56 ^c^	Day 84
1	PBS	PBS	PBS	1 × 10^3^ PfCSP-Tc/Pb Sporozoites
2	RTS,S 5 μg	RTS,S 5 μg	RTS,S 5 μg	
3	RTS,S 1.6 μg	RTS,S 1.6 μg	RTS,S 1.6 μg	
4	RTS,S 0.5 μg	RTS,S 0.5 μg	RTS,S 0.5 μg	

^a,b,c^ Homologous protein-in-adjuvant vaccine; ^a^: prime, ^b^: the 1st boost, and ^c^: the 2nd boost. PBS: phosphate buffer saline.

**Table 2 vaccines-12-01155-t002:** Immunization schedule for comparison of m8∆/AAV1 prime-boost vaccination regimen and low-dose RTS,S/AS01 (RTS,S) vaccine.

Mice	Group	Vaccine Administration				Challenge
		Day 0	Day 14	Day 28	Day 56	Day 84
BALB/c	1	PBS	N/A	PBS	PBS	1 × 10^3^ PfCSP-Tc/Pb Sporozoites
	2	RTS,S 0.5 μg ^a^	N/A	RTS,S 0.5 μg ^b^	RTS,S 0.5 μg ^c^	
	3	N/A	m8∆ 10^7^ PFU ^d^	N/A	AAV1 10^10^ vg ^e^	
C57BL/6	1	PBS	N/A	PBS	PBS	1 × 10^2^ PfCSP-Tc/Pb Sporozoites
	2	RTS,S 0.5 μg ^a^	N/A	RTS,S 0.5 μg ^b^	RTS,S 0.5 μg ^c^	
	3	N/A	m8∆ 10^7^ PFU ^d^	N/A	AAV1 10^10^ vg ^e^	

^a,b,c^ Homologous protein-in-adjuvant vaccine; ^a^: prime, ^b^: 1st boost, and ^c^: 2nd boost. ^d,e^ Heterologous prime-boost viral-vectored vaccine; ^d^: prime and ^e^: boost. N/A: not applicable, PBS: phosphate buffer saline, PFU: plaque forming unit, μg: microgram, vg: viral genome.

**Table 3 vaccines-12-01155-t003:** Transmission-blocking activity induced by passive transfer of sera from mice immunized with m8∆/AAV1 or RTS,S/AS01 (RTS,S) to naïve C57BL/6 mice.

Source of Sera	Group	Donor Serum	Oocysts Intensity (STDEV)	Infected Mosquitoes Prevalence (%)	TRA ^a,b^ (%)	TBA ^c,d^ (%)
BALB/c	G.1	Naïve	21.85 (29.04)	83.33		
	G.2	RTS,S	40.94 (56.64)	81.48	−87.37 ^ns^	2.22 ^ns^
	G.3	m8∆/AAV1	3.71 (6.17)	50.85	83.02 ****	38.98 ****
C57BL/6	G.4	Naïve	33.86 (39.79)	87.93		
	G.5	RTS,S	32.92 (43.15)	91.84	2.78 ^ns^	−4.45 ^ns^
	G.6	m8∆/AAV1	14.72 (22.56)	80.00	56.53 **	9.02 ^ns^

^a^ Transmission-reducing activity (TRA) was calculated for each group in comparison to the control group, and significant differences were assessed using a Mann–Whitney *U*-test (**** *p* < 0.0001, ** *p <* 0.01, ns; not significant). ^b^ A significant difference was found between the TRA of m8Δ/AAV1 and RTS,S vaccine groups of BALB/c (*p* < 0.0001) and C57BL/6 mice (*p* = 0.0027). ^c^ Transmission-blocking activity (TBA) was calculated for each group in comparison to the control group, and significant differences were assessed using a Fisher’s exact test (**** *p* < 0.0001, ns; not significant). ^d^ A significant difference was found in TBA between the m8Δ/AAV1 and RTS,S vaccine groups of BALB/c mice (*p* = 0.0007). No significant difference in TBA between the groups of C57BL/6 mice (*p* = 0.1055) was observed. STDEV: Standard deviation.

## Data Availability

The original contributions of this study are contained within the article or Appendix A. Data can be requested from the corresponding author.

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
