# Peer review of "A Head-to-Head Comparative Study of the Replication-Competent Vaccinia Virus and AAV1-Based Malaria Vaccine versus RTS,S/AS01 in Murine Models"

_vaccines, 2024, doi:10.3390/vaccines12101155_

Round 1
Reviewer 1 Report
Comments and Suggestions for Authors
Malaria is a life-threatening disease. Lack of effective vaccines has posted challenges in preventing malaria. This study directly compare a hetero prime-boost virus vaccine( m8Δ/AAV1) previously developed in the author’s lab with GSK’s RTS,S vaccine in malaria mouse models. This is a comprehensive study demonstrates the effectiveness of the viral-vectored vaccines which hold the potential to overcome the challenges associated with current protein-in-adjuvant vaccines. The manuscript meets the publication requirement.
Minor comments:
R21/Matrix-M has been the most effective malaria vaccine so far. And it is a subunit vaccine formulated with adjuvant, which can be easily manufactured, stored, and delivered. Though there were no data to compare m8Δ/AAV1 with R21/Matrix-M, it would be better if the authors include a brief discussion regrading this newly WHO endorsed vaccine.
Author Response
Comments 1:
Malaria is a life-threatening disease. Lack of effective vaccines has posted challenges in preventing malaria. This study directly compare a hetero prime-boost virus vaccine( m8Δ/AAV1) previously developed in the author’s lab with GSK’s RTS,S vaccine in malaria mouse models. This is a comprehensive study demonstrates the effectiveness of the viral-vectored vaccines which hold the potential to overcome the challenges associated with current protein-in-adjuvant vaccines. The manuscript meets the publication requirement.
Minor comments:
R21/Matrix-M has been the most effective malaria vaccine so far. And it is a subunit vaccine formulated with adjuvant, which can be easily manufactured, stored, and delivered. Though there were no data to compare m8Δ/AAV1 with R21/Matrix-M, it would be better if the authors include a brief discussion regrading this newly WHO endorsed vaccine.
Response 1:
The following sentence has been added.
It is worth noting the recent WHO endorsement of the R21/Matrix-M vaccine, which has demonstrated significant efficacy against malaria in phase III clinical trials. Developed as a subunit vaccine with an adjuvant, this vaccine is formulated to be easily manufactured, stored, and delivered. Although our study did not include direct comparative data between m8∆/AAV1 and R21/Matrix-M, both vaccines show promise in malaria control strategies. Future studies comparing m8∆/AAV1 with R21/Matrix-M could provide further insights into these vaccines' long-term protection, transmission-blocking efficacy, and practical deployment. (Please see the attachment: Lines 672-677).

Reviewer 2 Report
Comments and Suggestions for Authors
The present study by Shigeto Yoshida and coworkers evaluates the efficacy of a novel antimalarial vaccine in comparison to the RTS/S vaccine. The most important key findings of the manuscript are the protection by two boosts against different stages in the Plasmodium life cycle i.e. transmission and invasion into the hepatocytes in contrast to the RTS/S vaccine. Moreover, the vaccine protects against cerebral malaria. Interestingly, the m8Δ/AAV1 vaccine participates in a Th1 and Th2 response and stimulates different IgG subclasses compared to RTS/S achieving a broader immunity which might have an important translational effect.
The experiments are clearly presented and excellently performed and sound. The manuscript is very well structured and presents highly interesting and novel data.
Hence, I have some minor questions:
1. The construction of the double transgenic line is not completely clear. Why do the authors select a signal peptide and parts of the CSP coding sequence for the construct? This should be further explained.
2. I suggest that the abbreviations TRA and TBA etc. should be explained in a separate list.
3. It would be useful to introduce the differences between the RTS/S and the m8Δ/AAV1 vaccine on a molecular basis in a scheme at the beginning of the Results section or in the Introduction. Is it correct that the m8Δ/AAV1 contains a truncated version of CSP fused to a T cell epitope? What does the RTS/S vaccine contain?
Reviewer 3 Report
Comments and Suggestions for Authors
Authors developed a multistage Plasmodium falciparum vaccine (m8Δ/AAV1). The immunogenicity, protective efficacy and TB activity of LC16m8∆/AAV1 were detected by comparison with the established RTS,S/AS01 vaccine in murine models. m8∆/AAV1's prevented sporozoite invasion and onward transmission, superior to RTS,S. It is a very interesting work and provides important insights into the potential of the m8Δ/AAV1 vaccine to serve as an effective malaria vaccine. However, some questions need to resolve before publishing:
Line 35: please give the full name of TB.
Line 138: please provide more information of mice, such as sources and body weight.
In 2.3. Immunization section, why do you select the dosages of 5 µg, 1.6 µg, or 0.5 µg?
Please give ethical information in animal experimentation.
Line 458, please delete Table 2 or replace it with references.
The efficacy of both vaccines was 40% protection in C57BL/6 mice, lower than in BALB/c mice. Please explain them if it is possible. Why are C57BL/6 mice a more sensitive model than BALB/c mice for vaccination against malaria? Please give some references or explain it.
Other minors:
Ref. 10: 2023: Preprints with THE LANCET?
Ref. 38, 48, 52: the title
